# Near real-time surveillance of the SARS-CoV-2 epidemic with incomplete data

**Pablo M. De Salazar**[1]*, **Fred Lu**[2,3], **James A Hay**[1], **Diana Gómez-Barroso**[4,5], **Pablo Fernández-Navarro**[4,5], **Elena V Martínez**[5,6], **Jenaro Astray-Mochales**[7], **Rocío Amillategui**[4], **Ana García-Fulgueiras**[8], **Maria D Chirlaque**[8], **Alonso Sánchez-Migallón**[8], **Amparo Larrauri**[4,5], **María J Sierra**[6,9], **Marc Lipsitch**[1], **Fernando Simón**[5,6], **Mauricio Santillana**[1,2,3,10‡], **Miguel A Hernán**[11‡]

**1** Center for Communicable Disease Dynamics, Department of Epidemiology, Harvard TH Chan School of Public Health, Boston, Massachusetts, United States of america, **2** Machine Intelligence Lab, Boston Children's Hospital, Boston, Massachusetts, United States, **3** Computational Health Informatics Program, Boston Children's Hospital, Boston, Massachusetts, United States of america, **4** Centro Nacional de Epidemiología, Carlos III Health Institute, Madrid, Spain, **5** Consorcio de Investigación Biomédica en Red de Epidemiología y Salud Pública (CIBERESP), Madrid, Spain, **6** Centro de Coordinación de Alertas y Emergencias Sanitarias, Ministry of Health, Madrid, Spain, **7** Directorate-General for Public Health, Madrid General Health Authority, Madrid, Spain, **8** Department of Epidemiology, Regional Health Council, IMIB-Arrixaca, Murcia, Spain CIBER in Epidemiology and Public Health (CIBERESP), Madrid, Spain, **9** Consorcio de Investigación Biomédica en Red de Enfermedades Infecciosas (CIBERINF), Madrid, Spain, **10** Department of Pediatrics, Harvard Medical School, Harvard University, Boston, Massachusetts, United States of america, **11** CAUSALab, Department of Epidemiology and Department of Biostatistics, Harvard T. H. Chan School of Public Health, Boston, Massachusetts, United States of america

‡ These authors are joint senior authors on this work.
* pablom@hsph.harvard.edu

**Data Availability Statement:** The data that support the findings of the study was obtained from the Spanish System for Surveillance at the National Center of Epidemiology (RENAVE) through the

## Abstract

When responding to infectious disease outbreaks, rapid and accurate estimation of the epidemic trajectory is critical. However, two common data collection problems affect the reliability of the epidemiological data in real time: missing information on the time of first symptoms, and retrospective revision of historical information, including right censoring. Here, we propose an approach to construct epidemic curves in near real time that addresses these two challenges by 1) imputation of dates of symptom onset for reported cases using a dynamically-estimated "backward" reporting delay conditional distribution, and 2) adjustment for right censoring using the *NobBS* software package to nowcast cases by date of symptom onset. This process allows us to obtain an approximation of the time-varying reproduction number ($R_t$) in real time. We apply this approach to characterize the early SARS-CoV-2 outbreak in two Spanish regions between March and April 2020. We evaluate how these real-time estimates compare with more complete epidemiological data that became available later. We explore the impact of the different assumptions on the estimates, and compare our estimates with those obtained from commonly used surveillance approaches. Our framework can help improve accuracy, quantify uncertainty, and evaluate frequently unstated assumptions when recovering the epidemic curves from limited data obtained from public health systems in other locations.

Web platform SiViEs (System for Surveillance in Spain). The Spanish Ministry of Health has the policy not to provide publicly available line-listed data for confidentiality standards. Here, the authors provide code and anonymized line-list data (for the intermediate period of analysis) that allows reproduction of the results in the manuscript at: https://github.com/pdesalazar/Nowcasting_covid19_Spain. A version of the dataset used in the analysis but aggregated daily is publicly available at https://cnecovid.isciii.es/covid19/.

**Funding:** PMD was supported by the fellowship Ramón Areces Foundation. JAH was funded by the National Institute of General Medical Sciences, Award U54GM088558, and the National Institutes of Health Director's Early Independence, Award DP5-OD028145. ML was supported by the Morris-Singer Fund and by a subcontract from the Carnegie Mellon University under an award from the US Centers for Disease Control and Prevention, Award U01IP001121. MS was supported by the National Institute Of General Medical Sciences, Award R01GM130668-02. The funders had no role in study design, data collection and analysis, decision to publish, or preparation of the manuscript.

**Competing interests:** I have read the journal's policy and some of the co-authors of this manuscript have the following competing interests: ML discloses honoraria/consulting from Merck, Affinivax, Sanofi-Pasteur, Bristol Myers-Squibb, and Antigen Discovery; research funding (institutional) from Pfizer, and an unpaid scientific advice to Janssen, Astra-Zeneca, One Day Sooner, and Covaxx (United Biomedical). MS discloses having received institutional research support from Johnson and Johnson.The rest of co-authors declare no competing interest.

## Author summary

When surveillance systems cannot be repurposed quickly enough for novel infectious agents, missing information becomes a major challenge in monitoring the outbreak in real time. We propose a statistical approach that uses available data to construct the epidemic curves, which describe the number of individuals infected over time. We apply our 3-step approach to estimate these real-time epidemic curves during the early SARS-CoV-2 outbreak in Spain. We demonstrate that our approach, combined with the understanding of its limitations, can (a) provide useful information earlier and more reliably than conventional surveillance approaches, and (b) aid in the decision-making process towards outbreak mitigation in real-time.

## Introduction

Assessing the effectiveness of interventions during outbreaks requires real-time characterization of new infections. Epidemic curves, which describe the number of individuals infected over time, are frequently used to monitor the dynamics of an outbreak. Constructing these curves is particularly challenging for novel pathogens because testing protocols and surveillance systems may not be repurposed quickly enough.

Ideally, epidemic curves should document new infections based on the date of exposure to the infectious agent for each individual [1]. Because information on the date of exposure for each patient is usually unavailable, epidemic curves are often constructed based on the first detectable clinical event: onset of symptoms. In practice, however, detected cases are rarely documented at onset of symptoms. Rather, surveillance procedures tend to rely on confirmed diagnoses, which are typically reported with a delay of days or weeks after symptoms onset [2–4]. As a result, the number of cases based on the onset of symptoms on any given day is unknown until those cases are reported days or weeks later. This problem is sometimes referred to as "backfill bias", a term from economics [5] that was later applied to infectious disease tracking [6,7].

Reporting delays complicate timely decision-making [8]. Further, similar populations with different notification procedures may have different reporting delays [9] and the distribution of delays may change over time [10,11]. One way to address the problem of delayed notification is to statistically predict the number of cases with onset of symptoms today based on historical observations of how the number of cases on a given day were later revised to reflect updated information. This methodology is referred to as "nowcasting" [6,10]. A recently proposed approach to nowcasting, *NobBS* or Nowcasting by Bayesian Smoothing [10], complements the estimated delay distribution and historical data with the intrinsic autocorrelation from the transmission process. *NobBS* has been shown to perform better than previous nowcasting methods for different infectious diseases [10].

Nowcasting requires that the date of symptom onset is collected for all cases that are eventually reported. However, real-time surveillance systems cannot guarantee the ascertainment of the date of the clinical event for all reported cases, even if all cases are reported. Therefore, nowcasting the epidemic curve based on symptoms onset requires imputation of the missing dates of symptoms onset. The nowcast epidemic curve can then be used to estimate the time-varying reproductive number, that is, the number of secondary infections arising from a single infection on a particular day [12,13]. The estimation of the reproductive number also requires parametric assumptions on the generation interval (the time between a primary and a secondary infection).

Here, we present a three-step approach to estimate in near-real time the epidemic curve and the time-varying reproductive number ($R_t$) in the presence of reporting delays and incomplete data on the date of onset of symptoms. First, we impute the missing date of onset of symptoms data using historical distributions derived from reported line-list data. Second, we use *NobBS* to estimate case counts up to the present while adjusting for reporting delays. Third, we estimate the time-varying reproductive number using the nowcasted epidemic curve. We apply the approach to data reported during the early stages of the SARS-CoV-2 outbreak in two regions of Spain.

## Methods

### Ethics statement

The surveillance protocol was approved by the Inter-territorial Council of the Spanish National Health System. Although individual informed consent was not required, all data were pseudonymised to protect patient privacy and confidentiality. The study was also reviewed by the Institutional Review Board of the Harvard T.H. Chan School of Public Health, Boston, MA (US)

### Surveillance data

We applied our methodology to Madrid and Murcia, two regions of Spain with very different characteristics. Madrid has 6.7 million residents, is highly interconnected both nationally and internationally, has the highest population density and urbanicity of the country, is situated in the geographic (inland) center, and had a seroprevalence for SARS-CoV-2 of 11.5% at the end of the study period [14]. Murcia has 1.4 million residents, average connectivity, population density and urbanity, is geographically situated in the coastal periphery and had a seroprevalence of 1.6% at the end of the study period.

Each region reported daily counts of PCR-confirmed COVID-19 cases to the Spanish Ministry of Health [15] and individualized data on date of report (DOR) and, for a proportion of cases, the date of symptom onset (DOS) to the Spanish System for Surveillance at the National Center of Epidemiology (RENAVE) through the Web platform SiViEs (System for Epidemiologic Surveillance in Spain) [16].

We conducted the analyses in each region using cumulative data available at three overlapping periods in the outbreak: early analysis period when reported cases reached maximum counts (spanning March 1-March 27), intermediate analysis period shortly after the peak of the epidemic curve (March 1-April 9), and late analysis period when the epidemic curve was close to zero (March 1-April 16). We chose these three periods because their ending points correspond to distinct times when decisions about epidemic control were considered in Spain, and thus the limitations arising from the availability of data were especially relevant at those points. All analyses were implemented in R version 4.0.2.

Our approach has three steps.

### Step 1: Imputation of missing data

We imputed the missing DOS by randomly assigning values drawn from the distribution of reporting delay (the period between DOS and DOR), conditional on DOR, in individuals with known DOS. Of note, the reporting delay distribution conditional on the DOR is different from both the unconditional delay distribution (the distribution of all reporting delays) and the "forward" delay distribution conditional on the DOS (*if my DOS is today, how long do I wait until DOR?*). Inferring reporting delays from the unconditional delay distribution or the "forward" delay distribution conditional on the DOS is known to generate biased epidemic

curves [17]. We first assumed that missing DOS occurred at random with respect to symptom onset date, and that reporting delays conditional on DOR can be modeled over time $t$ and location $i$ as a negative binomial distribution with mean parameter $\mu_{i,t}$ and dispersion parameter $\theta_{i,t}$. Therefore, samples of the missing backward reporting delays $Y^m_{i,t}$ can be obtained from the parametric approximation of the observed backward reporting delays over the same time and location, $Y^o_{i,t}$, resulting in the following model:

$$Y^m_{i,t} \simeq Y^o_{i,t}$$

$$Y^o_{i,t} \sim NB(\mu^o_{i,t}, \theta^o_{i,t})$$

We dynamically estimated the parametric delay distribution at each day by regressing the backward delay distribution from all cases with available DOS pooled over a period of time $\tau$ comprising all dates between the day of imputation $d$ and a lag $u$ (i.e., $\tau = d-u,\ldots d$) and using maximum likelihood [9]. Using observations over $\tau$, instead of only using observations at $d$ allows us to increase the size of the observed delays at each location for the fitting step. However, to adjust for variation on the reporting patterns due to the day of the week (i.e., reduced reporting during weekends compared to weekdays) as well as different short-term dynamical trends (i.e. increasing/decreasing trend over a week), we modeled the mean delay conditional to the categorical predictor $w$ being the reporting date weekday or weekend:

$$\log(\mu^o_{i,\tau}) \sim \beta w^o_{i,\tau}$$

For the main analysis we pooled from a 3-days period when the number of available observations were 50 or more, and sequentially increased $\tau$ to 7 or 10 days if the total number of observations in $\tau$ were smaller than 30. When the number of observations included in the longest period of time (10 days) were less than 30, we simply imputed the missing delays by subtracting the observed mean delay from the DOR of each case with incomplete information. Last, for imputing the missing DOS at each reporting day $d$ and aiming to reliably estimating the uncertainty around the epidemic trends, we sampled from a randomly generated negative binomial distribution where $\mu^m_{i,d}$ and $\theta^m_{i,d}$ were modeled as having both a normal distribution with the mean set to the estimated $\hat{\mu}^o_{i,d}$ and $\hat{\theta}^o_{i,d}$, and the standard deviation set to the standard error of the mean. We resampled 100 times to generate 100 time series of cases with complete DOS-DOR for each region $i$. The sum of the observed and imputed cases became the total cases used for nowcasting. For further details on the imputation model see S1 Text.

We conducted several sensitivity analyses to explore the impact of the method's assumption on the estimates. First, we repeated the main model approach after masking DOS (hiding original data as missing values) in a random 10% and a random 40% sample of the cases with available reporting delay. Second, after masking DOS in a random 40%, we used two alternative models for imputing the missing reporting delays: a) using a single mean parameter $\mu_{i,d}$ over days and region (stronger assumption than the main approach) b) using a mean parameter $\mu_{i,d}$ and dispersion parameter $\theta_{i,d}$ estimated always from the distribution of the observed delays pooled from a 7-days period ($\tau$) instead of a 3-days period, where $\tau = d-6,\ldots d$ (stronger assumption than the main approach but weaker than a). Third, we masked DOS in a random 40% sample of cases and then imputed the missing DOS by subtracting the observed mean reporting delay from the reporting date for each case [17] (a procedure hereinafter referred to as backshifting). Last, we evaluated how deviations from the missing-at-random assumption would impact the imputation by randomly masking 20% cases with delays shorter than the median delay from that analysis period, or masking those cases with delays longer than the median.

### Step 2: Nowcasting the epidemic curve

After imputation, all reported cases have a DOS. However, a number of cases with DOS before day $t$ will be reported after day $t$. We therefore used *NobBS* [10] to nowcast the number of yet unreported cases at $t$. Briefly, *NobBS* uses historical information on the reporting delay to predict the number of not-yet-reported cases in the present using a log-linear model of the number of cases. We used the *NobBs* R package (v1.2), which compiles in *JAGS* using the *rjags* package (v4.10) to compute the number of cases reported with a particular delay. Cases are modeled as a negative binomial process. Further, the implementation of *NobBS* requires the specification of 1) a sliding window for the time-varying reporting delay, 2) a maximum delay allowed for the window, and 3) a set of priors of the negative binomial distribution parameters. We specified the moving window as the 75% of the total number of days in each period, the maximum delay as the maximum window minus 1 day, and weakly informative priors for the *NobBs* parameters. Detailed formulation of the nowcast model and prior distributions can be found in S1 Text. To assess uncertainty in the estimation, we produced a nowcast series with 10,000 posterior samples for each time point of the 100 imputed case count series in each region. We then pooled the samples and calculated the nowcast median and 2.5 and 97.5 percentiles for each day [18].

In sensitivity analyses, we a) used a fixed window of 4 weeks, b) varied the length of the window depending on the observed reporting delay over time, and c) generated the nowcasts using cases by report day and backshift by the mean reporting delay. Further, we evaluated the nowcast performance in more detail by assessing the changes on the nowcast estimates each day between March 25- April 8, 2020.

### Step 3: Estimation of time-varying reproductive number $R_t$

We estimated the time-varying reproduction number $R_t$ using two approaches by Wallinga and Teunis (WT) and Cori et al. (C) [13,19], both implemented in the R package *epiEstim* (v2.2.1). We used the nowcast estimates and a mean generation interval (the time between a primary and a secondary case infection) of 5 days with a standard deviation of 1.9 [20]. Details on the generation interval distribution can be found in S1 Text. We computed $R_t$ from the 10,000 *NobBS* samples to produce the mean and 95% credibility interval of the $R_t$ (for computational ease, we used a random sample of size 100 from the 10,000 samples since results did not materially change with a larger sample). Because WT and C use different forward and backward-looking computational approaches, respectively, a relative delay approximating the mean generation interval of WT relative to C is to be expected [17]. Further, downward biased WT estimates are expected for the last period of nowcast of length equivalent to the generation time, as they would lack sufficient data for computation.

In sensitivity analyses, we used available cases by DOS without imputation and nowcasting (i.e., without adjusting for missingness and censoring), observed cases by DOR with and without backshifting by the mean reported delay, and also used a longer generation interval (mean = 7.5, SD = 3.5) [21]. Last, we evaluated how significant changes in ascertainment (50% lower for the first 2 or 4 weeks of transmission) can impact the nowcast reconstruction and the $R_t$ estimates.

## Results

### Imputation of missing data

The proportion of missing DOS among reported cases varied by region and epidemic phase. In Madrid, the percentage of missingness was 53% of 32,723 reported cases in the early analysis period, 37% of 50,745 in the intermediate analysis period, and 16% of 56,057 in the late analysis period. The corresponding numbers in Murcia were 25% of 831, 23% of 1433, and 11% of

1602. The distribution of the reporting delay was also region-specific and changed over time (S1 Fig).

The first row of Fig 1 for each region shows the weekly counts of cases by DOR with observed and missing DOS. The second row of Fig 1 shows the epidemic curve by DOS after the median of imputed cases (grey) was added to the observed cases with available DOS (blue). As expected, the uncertainty of the imputation increases with the proportion of missingness. Compared with the main analysis, the simpler but less flexible negative binomial model showed some limitations when a high proportion of DOS were missing (S1 and S2 Texts and S2 Fig). Additionally, the curve was skewed when the assumption of missing-at-random was violated (S2 Text and S3 Fig).

## Nowcasting the epidemic curve

The second row of Fig 1 shows the epidemiologic curve after nowcasting (yellow line). Nowcasting reconstructed the epidemic curve more reliably than unadjusted case counts, either by DOR or DOS, even in earlier phases. This is more easily seen in Fig 2: nowcasted case counts in the intermediate period represent more accurately those estimated in the latest period, either compared with the curve of raw observed (not missing) case counts (B and E) or the curve modeled by mean backshift (C and F). However, the nowcasting procedure can be biased when there is a high proportion of missing data and sudden changes in the reported case numbers close to the end of the current observation (S3 Text).

Nowcast curves are smoother than curves of case counts by report date because of removal of noise related to the reporting process (such as weekday dependency). Also, the peak of case counts by symptom onset was several days earlier in the nowcast curves (around March 16 in both regions) than in the curves of reported cases (around March 25–26). The uncertainty in the nowcast increases with uncertainty in the imputation of DOS, as can be seen in the large uncertainty band in the early analysis.

The nowcast curve trends were relatively robust to small-to-moderate changes in missingness and different parameterizations of the *NobBS* function, such as the selection of the sliding window of analysis. However, the time of peak of case-counts can significantly vary with data availability and assumptions (S2 Text, S2 Fig). Further, major changes in ascertainment and/or reporting rates can substantially bias the nowcast estimates (S4 Text, S8 Fig)

## Estimation of the time-varying reproductive number

The third row of Fig 1 shows the $R_t$ estimates using the nowcasted curve and including uncertainty from previous steps. The precision of $R_t$ increased when more observed cases became available over time, as seen in the third row of Fig 1 for both regions. $R_t$ estimates computed using nowcasted curves show an earlier reduction of $R_t$ than those obtained from raw case counts by DOR as seen in Fig 3; further, $R_t$ estimates from case counts by DOR showed significantly higher values followed by a steeper slope for both approaches, and more noise when using Cori's approach compared to the estimates from the nowcasted curves. $R_t$ estimates obtained using observed case counts by DOS (i.e., without performing imputation and nowcasting) shifted in both locations towards lower values, particularly for the first periods of analysis, which in turn lead to a 6–9 days earlier estimates of the critical point where $R_t$ becomes <1; the estimates improved once more information was available for the last period (S4 Text, S7 and S8 Figs).

Though the WT and C approaches had similar trajectories, WT reached lower values earlier, with a delay between them approximating the mean generation time (5 days), the rationale being described previously [17]. This led to a consistent 4–6 days difference in the median estimated time for $R_t$ becoming <1, which can be seen in Fig 1 lower row for both regions. This

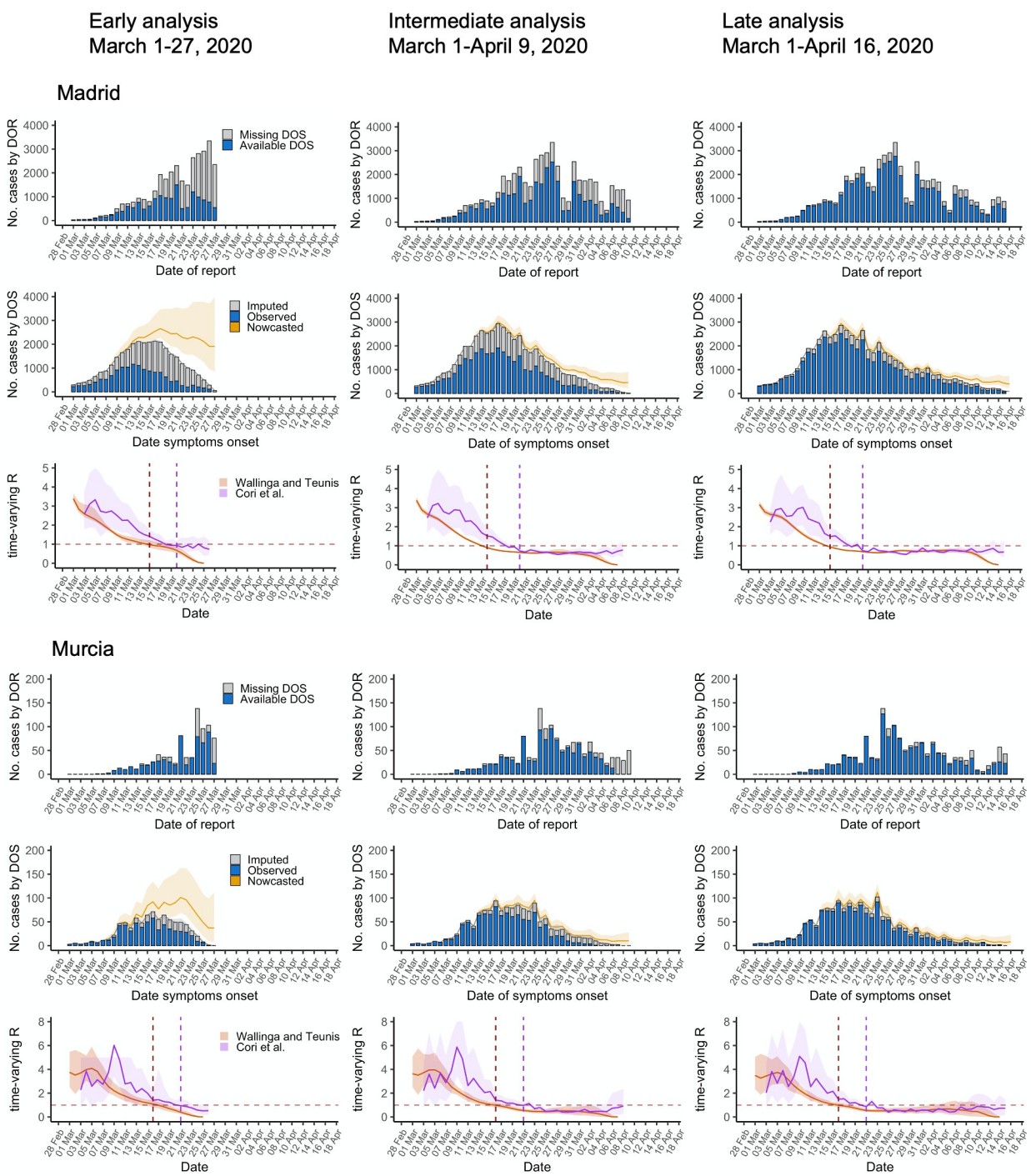

**Fig 1. Epidemic curves and reproductive numbers estimated using the data available during the early, intermediate, and late analysis of the initial SARS-CoV-2 outbreak in the regions of Madrid and Murcia, Spain, March 1-April 16, 2020.** DOS: date of onset of symptoms; DOR: date of report; Lines are median estimates, ribbons span 2.5 and 97.5 percentiles. Vertical lines indicate the day when $R_t < 1$ (red dashed line for WT, purple dashed line for C).

consistency was lost when using case counts by DOR, as seen in Fig 3. As expected, $R_t$ estimates using the WT method for the last days of analysis were biased downward, which precludes its use for the last period of length equal to the generation interval.

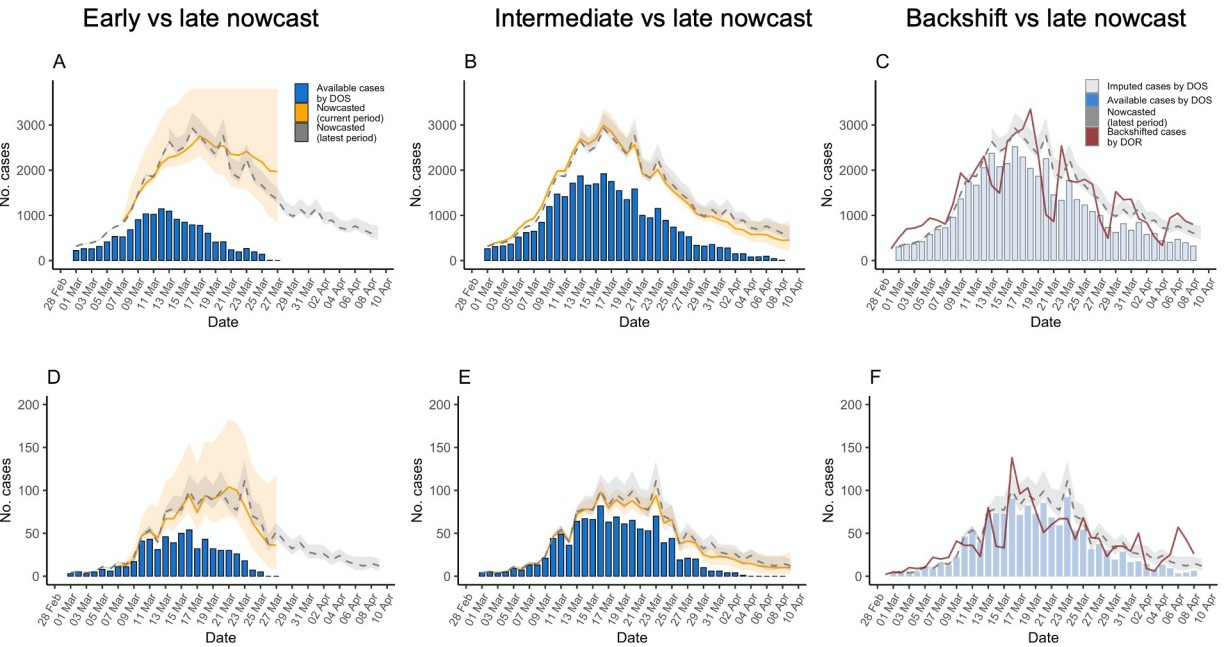

**Fig 2. Epidemic curves estimated using the data available during the early and intermediate analysis of the initial SARS-CoV-2 outbreak in the regions of Madrid and Murcia, Spain, March 1-April 9, 2020, and comparison with curves obtained in late period of analysis March 1-April 16 (dashed grey line and grey ribbon).** Showing nowcast estimates (orange) in Madrid and Murcia for the early (A and D) and intermediate (B and E) period analysis, observed cases with known date of onset of symptoms for the same period (blue columns), and for comparison with more complete data, those estimated in the late period of analysis (dashed grey line and ribbon); C and F showing cases by date of report back shifted by the mean delay (red line) together with nowcast estimates for the late period analysis and observed cases with known date of onset of symptoms (shadowed blue columns). DOS: date of onset of symptoms; DOR: date of report; Lines are median estimates, ribbons span 2.5 and 97.5 percentiles.

## Discussion

We proposed a three-step approach to characterize an outbreak in near real-time by adjusting for incomplete data and reporting delays. We applied this approach to the early SARS-CoV-2 outbreak in two regions of Spain. Our findings showed that a country-wide lockdown control was followed by a substantial decline in diagnosed cases shortly thereafter, around March 14–20 in Madrid and around March 17–23 in Murcia.

Nowcasted case counts were more accurate and consistent with true transmission than the unadjusted curve by DOS or DOR (as documented using complete data that became available after the study period). For example, our approach could identify SARS-CoV-2 epidemic control by lockdown in Spain almost a week earlier than when using reported cases by DOR.

Our approach has several limitations. First, its validity relies on the assumption that the date of symptoms onset is missing at random, that reporting delays can be approximated using a parametric approach, and that the available historical data are sufficient to parameterize the unknown reporting delay. These assumptions might be violated in many different ways depending on factors such as the accuracy, quality or procedures of the surveillance systems. However, our sensitivity analyses indicate that the overall trajectory of the epidemic curve was relatively robust to small-to-moderate departures from these assumptions. Second, our approach underperforms when little information is available for training the nowcasting algorithm, especially when good estimates of the reporting delay distribution cannot be obtained. This limitation might be particularly important a) in the very early stages of an emerging outbreak, when deficits among surveillance procedures, such as challenges in data submission to

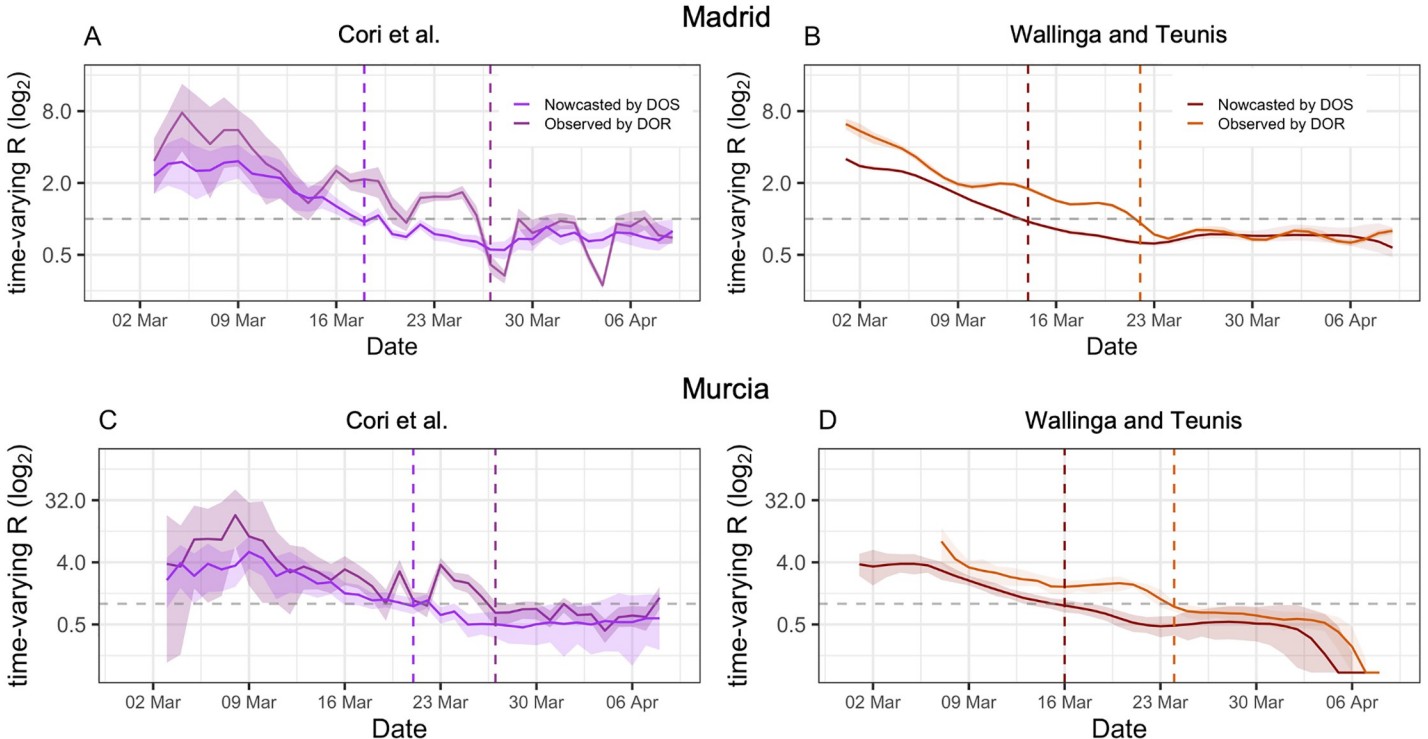

**Fig 3. Reproductive numbers estimated from nowcasted cases by DOS vs observed cases by DOR using Cori et al (A,C) and Wallinga and Teunis (B,D) approaches for Madrid and Murcia, March 1-April 09, 2020.** Showing the reproductive numbers estimated for Madrid (A,B) and Murcia (C,D). Estimates were computed using Cori et al (A,C) or Wallinga and Teunis (B,D) approaches and either nowcasted cases by DOS vs observed cases by DOR. A generation interval of mean 5 (1.9 SD) was used. Log2scale for the y-axis is used to facilitate visualization.

surveillance systems, may heavily impact the availability and consistency of data over time, and b) when the difference in the number of reported cases between consecutive days is substantially large, as for example, when there is a substantial reduction in case reporting during weekends. This limitation might preclude public health action based on day-to-day changes of the estimated number of infections; instead, changes in the number of infections during larger periods should be evaluated. Alternatively, more refined nowcasting models dealing with the forward reporting dynamics [22] could improve the reliability of the estimates. Third, our estimates were sensitive to the choice of imputation and nowcasting procedures when the date of symptoms onset was unknown for a high percentage of confirmed cases. Fourth, our approach requires that the degree of underreporting of cases is relatively constant over time. Major changes in case reporting will bias the $R_t$ estimates [23,24]. However, $R_t$ estimates remain unbiased if the proportion of unascertained or underreported observations remains time invariant [17,25]. All the previous limitations underscore that a good performance of the nowcasting approach can only be achieved by adequately specifying the model to account for the actual reporting process in the region that is analyzed. Future approaches including additional terms in the regression models can be explored aiming to better account for the reporting dynamics and the challenge of imputing under a small number of observations, while performing formal evaluation procedures can help to support selection of models with highest nowcasting accuracy [22]. Last, our estimates could be improved by reconstructing the epidemic curve by the date of infection rather than that of symptoms onset, though this would require more complex

methods given that the temporal delay from infection to symptom onset is much harder to characterize [17,26–29].

The overall findings of our work are consistent with the evaluation of a similar 3-step approach proposed recently [22] and analyses using synthetic data [17]. Nevertheless, we extend the analysis by focusing on, and illustrating, key aspects of the method and assumptions that support further adaptation of the approach to surveillance in other settings; by comparing different regions and periods of analysis; and by providing alternative models for reconstructing the epidemic curve and their evaluation while using existing computational tools.

Development of ready-to-use approaches for epidemic dynamics modelling help surveillance services to appropriately present data for efficient epidemic control, but understanding the limitations of the procedure and the impact of prespecified assumptions is critical for interpretation. Our approach provides a systematic analysis on key assumptions and implementation procedures frequently used to characterize emerging outbreaks. We propose a disease surveillance framework that acknowledges and adjusts for biases arising from real-world observational challenges, and is capable of providing objective, quantifiable, and systematic information, to aid the decision-making process in real-time outbreak mitigation efforts.

## Supporting information

**S1 Text. Additional information on model specification.**
(PDF)

**S2 Text. Sensitivity of the imputation step.**
(PDF)

**S3 Text. Sensitivity of the nowcasting step.**
(PDF)

**S4 Text. Sensitivity of the $R_t$ estimation step.**
(PDF)

**S1 Fig. Empirical distribution and approximated functions of the reporting delay conditional on report date in the regions of Madrid and Murcia, Spain, March 1-April 16, 2020.** The blue columns represent the true observed proportion. The grey line and ribbon represent the median and 95% uncertainty interval of the prediction under the fitted negative binomial distribution.
(PDF)

**S2 Fig.** Plotting the sum of imputed and observed epidemic curves (black line median, ribbon 95% CI) in the regions of Madrid and Murcia, Spain, March 1-April 16, 2020, estimated after A,F) randomly masking 10% of available reporting delays and using the main imputation approach, B,G) randomly masking 40% of available reporting delays and using the main imputation approach, C,H) randomly masking 40% of available reporting delays and estimating from a 7-day window of observations and allowing additional variance for the dispersion parameter D,I) randomly masking 40% of available reporting delays and using a single value for the mean in the negative binomial, and E,J) randomly masking 40% of available reporting delays and imputing by mean delay backshifting. Blue columns represent true observed case counts by day of symptoms onset.
(PDF)

**S3 Fig.** Plotting the sum of imputed and observed epidemic curves (black line median, ribbon 95% CI) in the regions of Madrid and Murcia, Spain, March 1-April 16, 2020, estimated after

A, B) randomly masking 20% of available reporting delays among cases with delays longer than the median and C, D) randomly masking 20% of available reporting delays among cases with delays shorter than the median. Blue columns represent true observed case counts by day of symptom onset.
(PDF)

**S4 Fig. Evaluation of the reconstructed epidemic curves at each day between March 25-April 8 using the data available during the intermediate period of analysis of the initial SARS-CoV-2 outbreak in the regions of Madrid and Murcia, Spain.** Nowcast estimates in Madrid and Murcia (orange lines are median estimates, ribbons span 2.5 and 97.5 percentiles, observed cases with known date of onset of symptoms for the late period of analysis (blue columns), cases with imputed date of onset of symptoms for the late period of analysis (grey columns) and nowcasted uncertainty range for the late period of analysis (grey ribbon); Mon: Monday, Tue: Tuesday, Wed. Wednesday, Thu.: Thursday, Fr.: Friday, Sat.: Saturday, Sun.: Sunday.
(PDF)

**S5 Fig. Epidemic curves estimated using alternative values for the *NobBS* sliding window and the data available during the early analysis of the initial SARS-CoV-2 outbreak in the regions of Madrid and Murcia, Spain, March 1–27, 2020, comparison with backshifted epidemic curves by report date, and curves obtained in late period of analysis March 1-April 16 using the main approach.** Nowcast estimates in the intermediate period of analysis (yellow lines represent the median and ribbons span the 2.5 and 97.5 percentiles) 1) using a fixed window of 28 days (A and E), 2) using a window that accounts for 75% of the delays available from the latest period of observations (B and F) 3) using a window that account for 99% of the observed delays (C and G); and 4) imputing through backshifting the report date by mean delay (D and H). Dashed lines and light grey ribbon represent nowcasted cases later in time using the main approach (late period of analysis). Faded blue columns represent observed cases by DOS, faded grey columns represent median imputed cases by DOS; dark grey ribbons (in A, B, C, D, E and F) represent observed plus imputed 2.5 and 97.5 percentiles.
(PDF)

**S6 Fig. Epidemic curves estimated using alternative values for the *NobBS* sliding window and the data available during the intermediate analysis of the initial SARS-CoV-2 outbreak in the regions of Madrid and Murcia, Spain, March 1-April 9, 2020, comparison with backshifted epidemic curves by report date, and curves obtained in late period of analysis March 1-April 16 using the main approach.** Showing nowcast estimates in the intermediate period of analysis (yellow lines represent the median and ribbons span the 2.5 and 97.5 percentiles) 1) using a fixed window of 28 days 2) using a window that accounts for 75% of the delays available from the latest period of observations (B and F) 3) using a window that account for 99% of the observed delays (C and G); and 4) imputing through backshifting the report date by mean delay (D and H). Dashed lines and light grey ribbon represent nowcasted cases later in time using the main approach (late period of analysis). Faded blue columns represent observed cases by DOS, faded grey columns represent median imputed cases by DOS; dark grey ribbons (in A,B,C, D, E and F) represent observed plus imputed 2.5 and 97.5 percentiles.
(PDF)

**S7 Fig. Reproductive numbers estimated comparing 2 different generation intervals on nowcasted curves and curves by report date using the data available during the intermediate analysis of the initial SARS-CoV-2 outbreak in the regions of Madrid and Murcia, Spain, March 1-April 9, 2020.** Showing sensitivity analysis of the $R_t$ estimates. Rows A and D

plots $R_t$ for WT (red) and C (purple) approaches using the nowcast estimates and a generation interval of mean 5 (1.9 SD). Estimates are also obtained using a longer generation interval of mean 7.5 (3.4 SD) shown in rows B and E; Rows C and F shows $R_t$ estimates using the shorter serial interval but calculated from the observed cases by date of report. Vertical lines indicate the day when $R_t < 1$ (red dashed line for WT, purple dashed line for C).
(PDF)

**S8 Fig. Reproductive numbers estimated using only available cases by DOS vs. nowcasted curves and curves by report date using the data available during the three periods of analysis the initial SARS-CoV-2 outbreak in the regions of Madrid and Murcia, Spain, March 1-April 9, 2020.** Lines are median estimates, ribbons span 2.5 and 97.5 percentiles. Vertical lines indicate the day when $R_t < 1$ ($R_t$ estimated from nowcasted curves are shown in red dashed line for WT, purple dashed line for C; $R_t$ estimated from available cases by DOS are shown in blue).
(PDF)

**S9 Fig.** Epidemic curves and $R_t$ estimates using the data available during the the intermediate analysis of the initial SARS-CoV-2 outbreak in the regions of Madrid and Murcia, Spain, March 1-April 9, 2020 (A,D and G,J) and comparison of the same estimate approach after randomly subtracting 50% of cases during the first 2 weeks (without including peak transmission, B,E and H,K) and during the first 4 weeks (including peak transmission, C,F and I,L). DOR: date of report; Lines are median estimates, ribbons span 2.5 and 97.5 percentiles. Vertical lines indicate the day when $R_t < 1$ (red dashed line for WT, purple dashed line for C).
(PDF)

## Acknowledgments

We thank Laura White and Rene Niehus for their technical assistance.

## Author Contributions

**Conceptualization:** Pablo M. De Salazar, Fred Lu, James A Hay, Amparo Larrauri, María J Sierra, Marc Lipsitch, Fernando Simón, Miguel A Hernán.

**Data curation:** Pablo M. De Salazar, Diana Gómez-Barroso, Pablo Fernández-Navarro, Elena V Martínez, Jenaro Astray-Mochales, Rocío Amillategui, Ana García-Fulgueiras, Maria D Chirlaque, Alonso Sánchez-Migallón.

**Formal analysis:** Pablo M. De Salazar, Fred Lu, James A Hay, Marc Lipsitch, Mauricio Santillana, Miguel A Hernán.

**Funding acquisition:** Pablo M. De Salazar.

**Investigation:** Pablo M. De Salazar, James A Hay, Diana Gómez-Barroso, Pablo Fernández-Navarro, Jenaro Astray-Mochales, Rocío Amillategui, Ana García-Fulgueiras, Maria D Chirlaque, Alonso Sánchez-Migallón, Amparo Larrauri, Fernando Simón, Miguel A Hernán.

**Methodology:** Pablo M. De Salazar, Fred Lu, Marc Lipsitch, Miguel A Hernán.

**Resources:** Diana Gómez-Barroso, Pablo Fernández-Navarro, Jenaro Astray-Mochales, Rocío Amillategui, Ana García-Fulgueiras, Maria D Chirlaque, Alonso Sánchez-Migallón, Amparo Larrauri, María J Sierra, Miguel A Hernán.

**Software:** Pablo M. De Salazar, Fred Lu.

**Supervision:** Marc Lipsitch, Fernando Simón, Mauricio Santillana, Miguel A Hernán.

**Validation:** Pablo M. De Salazar.

**Visualization:** Pablo M. De Salazar, Pablo Fernández-Navarro.

**Writing – original draft:** Pablo M. De Salazar, Fred Lu, James A Hay, Marc Lipsitch, Mauricio Santillana, Miguel A Hernán.

**Writing – review & editing:** Diana Gómez-Barroso, Pablo Fernández-Navarro, Elena V Martínez, Jenaro Astray-Mochales, Rocío Amillategui, Ana García-Fulgueiras, Maria D Chirlaque, Alonso Sánchez-Migallón, Amparo Larrauri, María J Sierra, Marc Lipsitch, Fernando Simón.

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
