## [Decision Letter · Decision Letter 0]

15 Feb 2021

Dear Dr. Martinez,

Thank you very much for submitting your manuscript "Near real-time surveillance of the SARS-CoV-2 epidemic with incomplete data" for consideration at PLOS Computational Biology.

As with all papers reviewed by the journal, your manuscript was reviewed by members of the editorial board and by several independent reviewers. In light of the reviews (below this email), we would like to invite the resubmission of a significantly-revised version that takes into account the reviewers' comments.

We cannot make any decision about publication until we have seen the revised manuscript and your response to the reviewers' comments. Your revised manuscript is also likely to be sent to reviewers for further evaluation.

Sincerely,

Benjamin Muir Althouse

Associate Editor

PLOS Computational Biology

Tom Britton

Deputy Editor

PLOS Computational Biology

Reviewer's Responses to Questions

**Comments to the Authors:**

Reviewer #1: This paper proposed a two-step approach to reconstruct the epidemic curve in nearly real time, firstly imputing the cases with missing symptom onset dates based on those with complete data and secondly nowcasting the epidemic curve to address the right censoring issue. This method addresses a very common issue faced by infectious disease epidemiologists worldwide, especially during the ongoing outbreak of COVID-19. The paper is well written and clearly presented. I have two comments that I hope the authors can address during revision.

First, since an important application of the method is to estimate Rt in real time, I would like to see a comparison of the Rt estimates with and without the proposed method. Currently, many reported Rt values for COVID-19 were based on the observed number of cases by symptom onset date (e.g., early papers based on data from Wuhan: PMID: 32275295 and 32674112). It will be good to make a direct comparison in this paper and discuss the implications for current COVID-19 surveillance.

Second, the authors pointed out one limitation of their approach is that it requires the ascertainment rate does not change significantly overtime. But the testing capacity in many countries has changed a lot since the outbreak. It will be good to use simulations to understand the impact of the violations of this assumption so that we can better interpret the results when applying their method.

Reviewer #2: The authors propose a three-step analysis of COVID-19 case reporting data for real-time monitoring of the epidemic situation and illustrate it based on the example of surveillance data from the first phase of the pandemic in Spain. The proposed analysis is based on three steps: 1) Imputation of missing dates of symptom onsets (DOS) based on the observed “backward” reporting delay distribution conditional on the date of report (DOR) from all cases with available DOS; 2) Nowcasting of the epidemic curve (number of cases with DOS on a given day) utilizing a previously published Bayesian model (Nowcasting by Bayesian Smoothing); 3) Estimation of the time-varying reproduction number R(t) based on two different approaches. The proposed analysis is illustrated in two Spanish regions at three different time points and results of the procedure for the first two time-points are evaluated based on retrospectively available data (at time point 3) to compare and evaluate different modeling choices.

Research on the adequate interpretation of COVID-19 case reporting numbers and real-time surveillance data is a timely and very important topic. Nowcasting can be a valuable analysis tool to adjust reported case numbers for reporting delays. The authors provide an interesting application of such a nowcasting approach, there are however several major aspects that require clarification/revision:

1) Missing references and novelty:

the proposed three-step analysis is not new, a similar approach of imputation of missing DOS, Bayesian nowcasting, and estimation of R(t) was already proposed and published in Guenther, et al. (2020). There it was shown that it is necessary to adjust for changes in the “forward” reporting delay distribution over time to avoid a bias in the estimation of the epidemic curve.

2) Missing methodological details and modeling choices:

2.1. Imputation: the authors model the reporting delay distribution given DOR based on a Negative Binomial model with varying mean per day and region and constant overdispersion parameter per region. Especially in times of low reporting numbers, modeling a single mean parameter per day might yield rather unstable results. On the other hand, the overdispersion of the NegBinom distribution might vary over time as well (as also described by the authors based on empirical data). Did the authors perform any (formal) model selection for the imputation model?

The imputation implies a missing at random assumption and the authors show in Supplemental section S1 that this imputation works quite well when the assumption is fulfilled (DOS missing for randomly selected individuals). In reality, the assumption might however be violated, e.g., if DOS is missing mainly because of pre-symptomatic cases. The current sensitivity analysis does not address this problem and this should at least be discussed. Furthermore, the possible extent of such a violation might be quantified in a a sensitivity analysis by setting the DOS of for cases with missing DOS to the DOR or e.g. DOR+2 in a sensitivity analysis.

2.2. Nowcasting: the description of the nowcasting model is very brief and hard for me to understand (even when additionally consulting the original NobBS publication). The selection and specification of the hyperparameters and priors is not motivated and incomplete. What was the maximum considered delay? Does the model assume a delay distribution that is independent of day t in the whole moving window? It is unclear to me what exactly the parameters alpha_t and beta_d as defined by the authors are and what assumptions the mentioned uniform priors imply. The notation and/or the prior do not correspond to the original publication of NobBS?

2.3. Estimation of R(t). The authors estimate R(t) based on the methods of Wallinga and Teunis and Corri et al. and note correctly that they are conceptually different. It would be helpful to provide information on the correct interpretation of the different estimated R(t)'s at a given timepoint t. Is there any advantage of one approach over the other?

Two further questions/remarks:

How was the generation time/serial interval distribution specified?

Because its definition is “forward looking”, the Wallinga and Teunis method is biased downwards for all days t for which no information on the number of disease onsets on day t+d is available for all number of days d that have a relevant probability mass in the generation time distribution (e.g., because t+d>current day T). For those days t it appears not to be meaningful to report the estimated R(t) based on WT.

3) Study design:

How and why did the authors choose the 3 specific time-points for showing the results of the nowcast? For the sake of illustration this might be a reasonable choice, but a thorough evaluation of the performance of the proposed approach should be made based on nowcasts for every day in the study period (i.e., comparison with “retrospective truth”). In this case coverage frequencies of prediction intervals can be investigated and systematic biases might be detectable. The performance of different model specifications might also be compared quantitatively, e.g. based on scoring rules.

4) Performance of analysis/nowcasting model:

the available information on the performance of the proposed analysis is insufficient. Especially around the first time-point in the Madrid region the estimated epidemic curve appears to be strongly biased upwards and the (retrospectively observable) decrease in the epidemic curve is not identified by nowcasting. This appears to be the most interesting time point with respect to real-time surveillance. To adequately judge the performance of the proposed analysis approach and its value for real-time surveillance a more formal quantitative evaluation would be necessary (see point 3.) Where does this strong bias at time point 1 come from, is it due to changes in the reporting delay distribution over time (e.g., increased reporting speed due to less cases/fewer workload for the health authorities)?

At what day is the peak in the epidemic curve in Madrid around March 15 identified based on the nowcasting?

5) Nowcasting with different moving window:

The presented results with respect to the specification of the moving window are difficult to follow. Based on figure S3 it appears like the maximum considered reporting delay was changed to 5 or 6 days in panel A/D, and for all days before T-5 or T-6 the so far available case counts are considered as final numbers. Based on the data presented in Figure S1 this appears not to be a reasonable analysis and it is therefore not possible to compare the effect of different moving windows for the estimation of the delay distribution. Furthermore, results for the first timepoint of analysis are missing and would be of main interest.

6) Unfortunately, there is a lack of data and code to reproduce the analysis. This makes it difficult to understand the analyses performed and hinders the reproducibility and application to data from different regions.

Literature

F. Günther, A. Bender, K. Katz, H. Küchenhoff, and M. Höhle. Nowcasting the COVID-19 pandemic in Bavaria. Biometrical Journal, 2020. https://doi.org/10.1002/bimj.202000112.

**Have all data underlying the figures and results presented in the manuscript been provided?**

Reviewer #1: Yes

Reviewer #2: **No: **The authors refer to a website (in Spanish) that provides aggregated Spanish COVID-19 case numbers. However, as far as I could see, it does not seem to be directly possible to obtain the person-specific or aggregated information on disease onset and reporting date, which are necessary for the proposed analyses of the manuscript. It is also not clear whether the historical data for the three different timepoints of analysis can be retrieved. Also, no code is available to reproduce the analyses. A code and data repository for reproducing and adapting the analyses would be desirable.

PLOS authors have the option to publish the peer review history of their article (what does this mean?). If published, this will include your full peer review and any attached files.

Reviewer #1: No

Reviewer #2: No
---

## [Decision Letter · Decision Letter 1]

3 May 2021

Dear Dr. Martinez,

Thank you very much for submitting your manuscript "Near real-time surveillance of the SARS-CoV-2 epidemic with incomplete data" for consideration at PLOS Computational Biology.

As with all papers reviewed by the journal, your manuscript was reviewed by members of the editorial board and by several independent reviewers. In light of the reviews (below this email), we would like to invite the resubmission of a significantly-revised version that takes into account the reviewers' comments.

We cannot make any decision about publication until we have seen the revised manuscript and your response to the reviewers' comments. Your revised manuscript is also likely to be sent to reviewers for further evaluation.

Sincerely,

Benjamin Muir Althouse

Associate Editor

PLOS Computational Biology

Tom Britton

Deputy Editor

PLOS Computational Biology

Reviewer's Responses to Questions

**Comments to the Authors:**

Reviewer #1: I think the authors misunderstand my first comment. The authors provided new results by applying WT and Cori's method on DOR for comparison. But what I am asking for is a comparison with direct estimate of Rt on DOS without imputation and nowcasting (as have been done in many studies during the pandemic). From Figure 1, we can see that imputation and nowcasting can change the epidemic curves substantially, but the impact on the Rt estimate is not clear. It will be good for the authors to show the difference and discuss the implications on epidemic control.

Reviewer #2: Dear Authors,

Thank you very much for providing a revised version of the manuscript and for the answers to my previous comments, through which several of my questions were answered. I have, however, some remaining questions/concerns that have to be addressed before I can recommend publication:

Despite the additional results and analyses, I am still not convinced about the performance and the added value of the proposed approach in the presented application scenarios, especially for the data in the first period in the Madrid region, which appears to me as the most important with respect to a (near) real-time assessment of the pandemic situation. This is due to the following reasons:

1) I am confused by the changes of the nowcast results for Madrid and Murcia at March, 27 (first period) compared to the initial submission. The results in Figure 1 and especially Fig 2A/D do not correspond to the results of Fig 2A/D in the initial submission of the manuscript. The bias in the results is not visible anymore but you did not state any changes with respect to the model or the utilized data in your response to the reviewer comments or the manuscript.

2) The results in the new sensitivity analysis (Fig S4) that incorporate more data (since they are based on the data from the second period restricted to what was available until March, 27) appear to be worse than the results presented in (the new) Fig 2A. This is very confusing and the introductory sentences for Supplementary Section S3 do currently not make sense to me. You state "As shown in Figure 1 in the main analysis, for the early period of analysis in Madrid, the nowcasting approach did not perform sufficiently well", but Figure 1/2A do now show quite accurate results. I am, however, not sure where they are actually coming from.

3) I am skeptical about your interpretation of the results of Figure S4. You state "We also observed that successive increases in the availability of data rapidly corrected the nowcast estimates as soon as March 28 (Figure S4 B and F)." The nowcast of March 29 (Figure S4-C) shows however, a strong bias into the opposite direction (i.e., an underestimation of the epidemic curve). The results from data up until march 27 and up until March 29 are qualitatively very different (and the situation was even worse with the data that was actually available until March 27 with more missing DOS). Altogether, in two out of the four days of the new sensitivity analysis, the 95%-PIs do not cover the retrospective "true" epidemic curve (based on information available at the third period) for several days close to "now". This raises the questions whether the approach would really be helpful in a real-time assessment of the current situation (and is capable "to aid decision-making process" as you state in the discussion) during the first period in Madrid and whether the uncertainty quantification of the method is adequate.

Based on these results, and despite the difficulties you have with the availability of the data, I would really recommend doing a proper quantitative evaluation of the performance of the nowcast model over a longer period of time to prove adequate performance. In addition, you should clarify the issues around Figs. 1 and 2 in your manuscript and discuss the problems with the nowcasting approach during the first period in more detail in the manuscript.

Some additional comments with respect to the imputation of missing DOS:

Based on the description of the imputation model in the manuscript, I still do not completely understand what the main model for imputation is. You write: "We assumed that missing DOS occur at random with respect to symptom onset

date, and that reporting delays conditional on DOR can be modeled over time t and location

i as a negative binomial distribution with mean parameter \\mu_{i,t} and dispersion parameter \\theta_i

estimated using maximum likelihood [9]."

Based on the code from github, it looks like you utilized a model

log(\\mu_{i,t}) = \\beta_{0,i} + \\beta_{1,i}*t

i.e., you assumed a linear time-trend for modeling the (log)-expectation of the reporting delay. This seems to be a rather unflexible model, but might be flexible enough for your data. However, I am not sure if you really used this model as you later say with respect to the sensitivity analyses:

"mean parameter mu_{i,t} and dispersion parameter theta_i,t [were] estimated from the distribution of observed delays pooled from t − tau to t (weaker assumption than the main approach with \\tau=7)".

I am somewhat confused what the final "main approach" is, please clarify this and specify the model using a clear description or formula!

For incorporating the imputation uncertainty into the nowcast you write that you "resampled 100 times to generate 100 time series of cases with complete DOS-DOR for each region i, allowing \\mu_{i,t} and \\theta_i to vary

randomly under a normal distribution with the standard deviation \\sigma set to the sampling error." I am not sure what you mean by "sampling error". I assume that you sample from the negative binomial distribution with parameters coming from draws of the normal distribution with parameters \\hat{mu} and \\hat{SE} of the corresponding parameters from the imputation model? It would then probably be better to replace "Sampling error" by "standard error" and also specify the expectation of the normal distribution you are sampling from.

Additional general comments:

- What do the vertical lines in Figure 3BD indicate? They do not seem to correspond to the time-points when R(t) crosses 1. Furthermore, you might consider plotting R(t) on a log-scale as this better describes the multiplicative interpretation of the reproductive number (and increases readability of your figures with very high values from the raw case counts).

- In the discussion, you mention that "Our findings showed that a country-wide lockdown control led to a substantial decline in cases shortly thereafter, around March 14-20 in Madrid and around March 16-21 in Murcia." I think that such causal statements are not covered by the presented analyses and it would be better to speak of a temporal association. I would also refrain from comparing the nowcasting results and the curve of case counts with respect to their consisteny with the date of the country-wide lockdown in the Results section.

- The description of the first-order Random walk in Section S1 seems to include an typo. Expectation of the normal distribution should probably be \\alpha_{t-1}

**Have the authors made all data and (if applicable) computational code underlying the findings in their manuscript fully available?**

Reviewer #1: Yes

Reviewer #2: **No: **Due to data confidentiality reasons only an anonymized line-list data is provided in addition to code that enables reproduction/mimicking of parts of the presented analyses.

PLOS authors have the option to publish the peer review history of their article (what does this mean?). If published, this will include your full peer review and any attached files.

Reviewer #1: No

Reviewer #2: No
---

## [Decision Letter · Decision Letter 2]

4 Aug 2021

Dear Dr. Martinez,

Thank you very much for submitting your manuscript "Near real-time surveillance of the SARS-CoV-2 epidemic with incomplete data" for consideration at PLOS Computational Biology.

As with all papers reviewed by the journal, your manuscript was reviewed by members of the editorial board and by several independent reviewers. In light of the reviews (below this email), we would like to invite the resubmission of a significantly-revised version that takes into account the reviewers' comments.

Please explicitly address all of reviewer 2's comments; more fully justify the novelty of this approach; and while you explore the limitations and potential problems of you analysis, you do not offer any solutions or suggestions on how to deal with such uncertainties in an actual applied analysis scenario. Address this as well, please.

We cannot make any decision about publication until we have seen the revised manuscript and your response to the reviewers' comments. Your revised manuscript is also likely to be sent to reviewers for further evaluation.

Sincerely,

Benjamin Muir Althouse

Associate Editor

PLOS Computational Biology

Tom Britton

Deputy Editor

PLOS Computational Biology

Please explicitly address all of reviewer 2's comments; more fully justify the novelty of this approach; and while you explore the limitations and potential problems of you analysis, you do not offer any solutions or suggestions on how to deal with such uncertainties in an actual applied analysis scenario. Address this as well, please.

Reviewer's Responses to Questions

**Comments to the Authors:**

Reviewer #1: All my comments have been addressed.

Reviewer #2: Dear authors,

I still have some major comments and remarks w.r.t to the current form of the manuscript that should be addressed:

Imputation model

Thanks for the clarifications with respect to the imputation model(s) in your last version of the manuscript. Based on these, I have some comments:

A) General description of Imputation model:

1. The formulation of the imputation model based on the formulas eq1-eq3 in S1 continues to be unclear, Eq 2 and Eq 3 seem to erroneous. In a way, they imply that the parameters are supposed to follow an additional model (as e.g., in a Bayesian hierarchical model) but this seems not to be the case based on the written text. Furthermore, it does not make sense that mu_it follows a normal distribution with expectation mu_it (i.e., itself) and why lamda_i,t is supposed to follow a normal distribution with expectation mu_i,t. Also it does not make sense that the variance parameters/standard deviation is shared between mu_i,t and theta_it.

2. From what I understand from the written text the backward delay distribution for reporting day t, t=1,…T, is just estimated by ML and you estimate the expectation and overdispersion parameter of a negative binomial distribution, where the data correspond to the (observed) delays of all individuals with reporting date t-1, t, and t+1. In this case, the parameters do not have any distribution but are assumed to be fixed, and only their estimates hat(mu) and hat(theta) have an approximate distribution/associated standard error based on ML-theory.

3. You then perform the imputation by sampling parameters of a negative binomial distribution based on the (marginal) approximate normal distribution of the estimated parameters hat(mu) and hat(theta) and sampling reporting delays for each individual from the corresponding distribution.

Please provide a valid description of your mathematical models!

B) General comments w.r.t. to the model (in real-time surveillance) as I understood it:

1. Estimating the reporting delay distribution for day t =1, …, T based on the days t-1 and t+1 appears to be questionable out of two reasons:

First, in real-time surveillance the data for T+1 is not available (where T is the current date, I.e., “now”). Second, infectious disease reporting data usually follows strong weekly patterns, and the authors mention this also for the Spanish data. This translates also to the backward delay distribution (e.g., if Sundays have few reported cases, the (average) reporting delays should be bigger on Mondays). It might therefore be problematic to estimate the reporting delay distribution for, e.g., a Sunday by aggregating over the collected data from Saturday-Monday, or for a Tuesday by aggregating over Monday-Wednesday. In future work you might just consider modelling the backward reporting delay distribution based on, e.g., a parametrical statistical (regression) model where you can account, e.g., for changes over time, week-day effects, etc. It is then also possible to perform a formal model statistical model evaluation/selection. You could add this option to the discussion.

2. The different approaches to model the reporting delay distribution for (3 consecutive) days with smaller than 50 reported cases compared to days with more than 50 reported cases raises some questions (this seems to be mainly relevant for Murcia). You write that theta is set to 1 for such days, and that this implies that P(d=a)>P(D=b), for all a<b. e.g.="" imply="" p="" that="" then="" this="" would="">P(D=1). This seems to be a questionable assumption, as reporting delay distributions usually have a peak of probability mass at a specific delay >0, as also seen in the data (Figure S1). Furthermore, I am irritated by the estimated delay distribution for Murcia for March 8 (Figure S1). The reported cases from Match 7-March 9 seem to be <50 (Figure 1), but the estimated delay distribution seems to have biggest probability mass for a delay of d=2. This seems to be inconsistent with the described imputation model. I do not think that this aspect plays a major role for the general results, but presented results should be consistent with the described modelling approach! Of note: A more general statistical model could also help with the problems arising on days with only few observations.

C) Reporting of the results:

- Caption in Figure S2 C,H seems to be wrong, the model is not more flexible than the revised main model.

D) Code:

The code in the GitHub repository page does still not correspond to the imputation model you are describing in the manuscript (as of July, 27, 2021)!

Nowcasting:

- Thanks for showing the results of your nowcasting method for additional days, this helps to judge the performance of the model better and the finding of a bias due to short-term changes in reporting activities on weekends is interesting and important to discuss.

- I agree with the conclusion that such biases complicate the interpretation of daily results, and I think it is good that this aspect is now being discussed. Your formulation “when there are large daily changes in the reporting case counts, such as that observed related to weekends, which might preclude public health action based on daily estimates” is however, a bit vague and seems to be grammatically wrong.

- In addition, it might be worth mentioning that consistent patterns in the reporting activity can also be considered in more refined nowcasting-models. This can, e.g., be done by letting the probability of being reported on day t+d (d=1, …, D) for an individual with disease onset at day t vary based on the weekday of day t+d and estimating those weekday effects within the nowcasting model. In the NobBS model this would correspond to modelling beta_td instead of only beta_d, e.g., based on a parametrical model that accounts for weekday changes.

- I think you should mention in the discussion that a good performance of a nowcasting approach can only be achieved when the model for estimating the reporting delays is adequately specified to account for the actual reporting process in the region that is analysed.

- Minor comment: In supp-note S3 you write “As shown in Figure 1 in the main analysis”: This should probably refer to Figure 2.

Estimation of R(t), new Figure 3:

- please add telling headings to the sub-figures and a legend to the plots. Use unique or reasonably shared colours for the different estimates of R(t), it is confusing that there are two blue lines

- why does the Walinga-Teunis based estimate for Murcia based on DOR (?, blue line in the new Fig 3D) ends at April 4? In addition, I find it surprising that the two estimates in 3D are so similar (until around March 25), do you have any explanation for this? Is this sub-figure and/or the caption correct or is this the estimate based on all available DOS without imputation and nowcasting?

Discussion

- You mention the central limitation of the case reporting data of a potentially time-varying dark-figure in reported cases and write “However, Rt estimates remain unbiased if the proportion of incomplete observations remains time invariant [17,25], as it is likely the case in our analysis.“ It is, however, not clear why the proportion of underreporting is “likely time invariant”. This should be either motivated more clearly or removed from the discussion. In Supp. Note 4 you write that “a 50% lower ascertainment occurring during the initial period of analysis […] is a plausible change, especially for the first weeks of transmission of COVID-19”. In addition, the wording “proportion of incomplete observations” appears to be unclear in the context of nowcasting. From my understanding the issue is about the proportion of missing observations/cases (underreporting) and not incomplete observations (i.e., cases without disease onset information).

General comments

A) You should revise the entire manuscript and supplementary texts for clarity of wording, correct use of statistical terms, and accurate definition of technical terms and mathematical/statistical variables. Please ensure that the captions of the figures are correct and self-explanatory, that colour choices are consistent and comprehensible, and that each figure has a legend. Besides the aspects I mentioned already above (e.g., mathematical model for reporting delays and various formulations), I found the following formulations in the current version of the main text to be unclear:

Methods, Step 1: Imputation of missing data:

- Despite my comment on the last version of the manuscript, you continue to use the term “sampling error” wrong. You write: “We resampled 100 times to generate 100 time series of cases with complete DOS-DOR for each region , allowing mu_it and sigma_it to vary randomly under a normal distribution with mean set to their estimated values, and standard deviation set to the sampling error”. The sampling error is usually defined as “the difference between a sample statistic used to estimate a population parameter and the actual but unknown value of the parameter”. This error cannot be estimated base on a single dataset. I think you should replace this by the “standard error of the estimate”.

- For the third imputation model in the sensitivity analysis, you write: “Third, we masked DOS in a random 40% of cases and then imputed DOS for all cases reported at any given t and then imputed the same date for all cases reported at any given t as the difference between the observed mean delay and the reported date”. This is unclear and the formulation “difference between the observed mean delay and the reported date” does not really make sense. You could describe this approach more clearly as an imputation by “subtracting the observed mean reporting delay from the reporting date for each case”. Here you might also define the term “backshift” that you are using later in the manuscript.

Methods, Step 2: Nowcasting the epidemic curve

- Please define your mathematical notation! What does lower-case t stand for? In the previous paragraph on imputation, you used it for each day in the whole analysis period. For describing the problem of nowcasting, it should probably stand for the last day in the observation period, this is often referred to as capital T.

Results, Nowcasting the epidemic curve:

- You write “However, when there is a high proportion of missing data the nowcasting procedure is sensitive to large changes in reported cases”. It is not clear what this means. Being sensitive to large changes is not necessarily something bad. I guess you mean that that large and sudden changes in reported case numbers close to the end of the current observation period (e.g., due to an unmodelled change in reporting activity during weekends), can bias/distort the estimates of your nowcasting procedure.

- Directly in the following you write “the bias decreases as more data becomes available”. Here it is not clear what bias you are referring to and when it decreases. After seeing more data from consecutive days?

- In the last sentence you write: “Further, the occurrence of major changes in ascertainment rates relative to the true epidemic trends might bias the nowcast estimates”. I am not exactly sure what you mean by “relative to the true epidemic trends”.

Results, Estimation of R(t)

- “The precision of Rt increased over time when more information became available”. Please be more precise what you mean by more information. Is it more reported disease onset dates and less imputation or just a bigger number of cases (higher incidence)? Or both?

- Please add a legend to Fig 3!

B) Please update the GitHub repository with the code that you are actually using in your analysis, currently it seems to be outdated and your data availability statement is therefore not correct!</b.>

**Have the authors made all data and (if applicable) computational code underlying the findings in their manuscript fully available?**

Reviewer #1: Yes

Reviewer #2: **No: **Available code does not match described methods and adresses only parts of the analysis.

PLOS authors have the option to publish the peer review history of their article (what does this mean?). If published, this will include your full peer review and any attached files.

Reviewer #1: No

Reviewer #2: No
---

## [Editor Report · Decision Letter 3]

24 Feb 2022

Dear Dr. Martinez,

We are pleased to inform you that your manuscript 'Near real-time surveillance of the SARS-CoV-2 epidemic with incomplete data' has been provisionally accepted for publication in PLOS Computational Biology.

Best regards,

Benjamin Althouse

Associate Editor

PLOS Computational Biology

Tom Britton

Deputy Editor

PLOS Computational Biology

Please make sure that the symbols appear as they should -- the version I have has boxes in place of many symbols.

---

## [Editor Report · Acceptance letter]

28 Mar 2022

PCOMPBIOL-D-20-02097R3 

Near real-time surveillance of the SARS-CoV-2 epidemic with incomplete data

Dear Dr De Salazar,

I am pleased to inform you that your manuscript has been formally accepted for publication in PLOS Computational Biology. Your manuscript is now with our production department and you will be notified of the publication date in due course.

With kind regards,

Zsofia Freund
